# SERCA Silencing Alleviates Aß_(1-42)_-Induced Toxicity in a *C. elegans* Model

**DOI:** 10.3390/ijms26189126

**Published:** 2025-09-18

**Authors:** Elena Caldero-Escudero, Silvia Romero-Sanz, Pilar Álvarez-Illera, Silvia Fernandez-Martinez, Sergio De la Fuente, Paloma García-Casas, Rosalba I. Fonteriz, Mayte Montero, Javier Alvarez, Jaime Santo-Domingo

**Affiliations:** 1Department de Biochemistry, Molecular Biology and Physiology, Faculty of Medicine, University of Valladolid (UVA), Ramón y Cajal, 7, E-47005 Valladolid, Spain; elena.caldero@uva.es (E.C.-E.); silvia.romero.sanz@uva.es (S.R.-S.); mariapilar.alvarez.illera@uva.es (P.Á.-I.); sergio.delafuente@uva.es (S.D.l.F.); paloma.garcia@uva.es (P.G.-C.); rosalba.fonteriz@uva.es (R.I.F.); mmontero@uva.es (M.M.); javier.alvarez.martin@uva.es (J.A.); 2Institute of Biomedicine and Molecular Genetics (IBGM), Higher Council for Scientific Research (CSIC)—University of Valladolid (UVA), Sanz y Fores, 3, E-47003 Valladolid, Spain; silviapatricia.fernandez@uva.es

**Keywords:** *C. elegans*, SERCA, *sca-1*, Alzheimer’s disease, beta-amyloid, mitochondria, ER Ca^2+^ signaling

## Abstract

The Sarco Endoplasmic Reticulum Ca^2+^-ATPase (SERCA) pumps cytosolic Ca^2+^ into the endoplasmic reticulum lumen (ER) to maintain cytosolic and ER Ca^2+^ levels under physiological conditions. Previous reports suggest that cellular Ca^2+^ homeostasis is compromised in Alzheimer’s Disease (AD) and that SERCA activity can modulate the phenotype of AD mouse models. Here, we used a *C. elegans* strain that overexpresses the most toxic human ß-amyloid peptide (Aß_(1-42)_) in body-wall muscle cells to study the effects of SERCA (*sca-1*) silencing on Aß_(1-42)_-induced body-wall muscle dysfunction. *sca-1* knockdown reduced the percentage of paralyzed worms, improved locomotion in free-mobility assays, and restored pharynx pumping in Aß_(1-42)_-overexpressing worms. At the cellular level, *sca-1* silencing partially prevented Aß_(1-42)_-induced exacerbated mitochondrial respiration and mitochondrial ROS production and restored mitochondrial organization around sarcomeres. *sca-1* knockdown reduced the number and size of Aß_(1-42)_ aggregates in body–wall muscle cells and prevented the formation of Aß_(1-42)_ oligomers. Aß_(1-42)_ expression induced a slower kinetics of spontaneous cytosolic Ca^2+^ transients in muscle cells and *sca-1* partially restored these changes. We propose that partial *sca-1* loss of function prevents the toxicity associated with beta-amyloid accumulation by reducing the formation of Aß_(1-42)_ oligomers and improving mitochondrial function, in a mechanism that requires remodeling of cytosolic Ca^2+^ dynamics and partial ER Ca^2+^ depletion.

## 1. Introduction

Alzheimer’s disease (AD) is a neurodegenerative disorder characterized by the progressive decline of cognitive functions and dementia [1]. Accumulation of aggregated beta-amyloid (Aβ) peptides in the form of plaques is a characteristic pathological feature of AD [2]. The Aβ hypothesis proposes that the accumulation of the Aβ-peptide in the brain is a central event in the development and progression of the disease rather than a passive readout of the disease evolution [3]. The hypothesis suggests that Aβ-peptide aggregates trigger cellular toxicity by several mechanisms, leading to synapse and neuronal loss, which ultimately produce the associated clinical symptoms [4]. Importantly, the initiation of the Aβ accumulation occurs decades before the onset of the symptoms and upstream to other pathophysiological hallmarks [5]. This evidence suggests a wide temporal therapeutic window to prevent the toxic actions of Aβ-peptide accumulation. Therefore, the identification of new genes and cellular processes reducing Aβ toxicity or boosting Aβ clearance is a promising strategy to discover new targets with disease-modifying properties.

Loss of cellular Ca^2+^ homeostasis has been widely reported as a cellular hallmark of AD [6]. The mechanisms underlying this phenomenon are diverse and include the formation of Aβ cation-selective ion channels [2] and the regulation of different key components of the Ca^2+^ signaling machinery by subunits of the γ-secretase complex and its substrate APP [7,8]. Several studies converge on the idea that the maintenance of the endoplasmic reticulum (ER) Ca^2+^ homeostasis, particularly the activity of the sarco-endoplasmic reticulum Ca^2+^-ATPase (SERCA), might be compromised in AD. In mammalian systems, lack of presenilins (PS1/2) have been shown to increase ER Ca^2+^ leakage [9]. SERCA may partially mediate this effect, as PS1 interacts physically with SERCA and both PS1/2 loss of function and overexpression of mutants associated with increased susceptibility to develop Familial Alzheimer’s Disease (FAD) reduced SERCA activity [10,11,12]. In addition, alterations in Ca^2+^ signaling also impact on APP maturation and proteolytic processing [13]. Specifically, SERCA activity has been shown to repress Aβ production [10]. More recently, pharmacological activators of SERCA such as CDN-1163, NCD-1173, and NDC-9009 have shown protective effects in murine models of AD [14,15,16]. Thus, compiling evidence in mammalian systems suggests that SERCA activity is a potential therapeutic target with disease-modifying properties.

*C. elegans* is also a valuable organism for modeling AD and studying the contribution of the Ca^2+^ signaling machinery to AD pathogenesis [17,18]. The *C. elegans* genome contains *sca-1*, a highly conserved orthologue of the mammalian SERCA [19]. *sca-1* is primarily expressed in contractile tissues, including body-wall muscle cells [19]. Lack of *sca-1* results in muscle dysfunction and in early larval or embryonic lethality [19], indicating that its activity is required for embryonic development. However, we have previously reported that both pharmacological inhibition and RNAi-induced silencing of *sca-1* in post-developed individuals promote health-span and longevity, suggesting that submaximal inhibition of *sca-1* triggers adaptations that improve organism survival in adult worms [20,21].

In the context of neurodegeneration paradigms in *C. elegans*, the role of *sca-1* remains under discussion. Several studies reported that inhibition of *sca-1* activity with thapsigargin potentiated neurodegeneration [22,23]. Conversely, *sca-1* inhibition with CPA in an α-synuclein overexpression model in dopaminergic neurons and *sca-1* inhibition with thapsigargin in a model of Aβ overexpression in glutamatergic neurons prevented neurodegeneration [24,25]. These results indicate that *sca-1* inhibition might be protective in models of toxic protein aggregation, although the underlying mechanisms are largely unknown. To obtain new insights into the influence of SERCA activity in the pathogenesis of AD, here we explore for the first time the impact of RNAi-driven *sca-1* silencing on Aβ-induced toxicity in a *C. elegans* model that overexpresses the most amyloidogenic version of the human Aβ_(1-42),_ in body-wall muscle cells.

## 2. Results

### 2.1. Sca-1 Silencing Prevents Paralysis and Enhances Mobility in Aβ_(1-42)_-Overexpressing Worms

Overexpression of human Aß_(1-42)_-peptide in body wall muscle cells of *C. elegans* induces the formation of Aβ aggregates and results in a temperature-sensitive paralysis phenotype when shifted from 20 °C to 25 °C [26]. To assess the impact of *sca-1* (SERCA) on Aß_(1-42)_-induced toxicity, we compared the percentage of paralyzed worms growing on plates containing *sca-1* RNAi-expressing bacteria to those growing on plates with an empty-targeted RNAi bacteria (L4440). RNAi-mediated knockdown of *sca-1* led to a 50% reduction in *sca-1* mRNA levels in the GMC101 strain (Figure 1D), confirming efficient gene silencing. As expected, following the temperature shift, control worms overexpressing Aß_(1-42)_ exhibited a progressive paralysis phenotype with aging (Figure 1A–C). In contrast, *sca-1* knockdown nematodes demonstrated a significant reduction in paralysis at days 4 and 8 of adulthood, especially with 100% RNAi (Figure 1A–C). Mild *sca-1* silencing has been shown to promote an increase in the lifespan of wild-type worms [20]. We thus explored whether mild *sca-1* silencing also counteracts Aß_(1-42)_-induced toxicity. However, nematodes fed with only 10% *sca-1* RNAi-expressing bacteria did not display any impact on paralysis at day 4 and just a minor, but significant improvement was measured at 8 of adulthood (Figure 1A–C).

To further characterize the impact of *sca-1* silencing on Aß_(1-42)_-induced body-wall muscle dysfunction, we measured the impact of *sca-1* silencing on locomotion parameters through free-mobility assays. Locomotion parameters measured included distance traveled and speed of movement, either as maximum speed, average speed, or speed measured in body lengths per second (BLPS) to account for differences in worm size. As previously reported [26], our results show that Aß_(1-42)_-overexpressing worms exposed to temperature challenge significantly reduced all mobility parameters during aging compared to N2 control worms (Figure 1E,G,I,K). Nematodes fed with 100% *sca-1* RNAi-expressing bacteria partially restored length traveled (Figure 1E,F), average speed (Figure 1G,H), maximum speed (Figure 1I,J) and body lengths per second (Figure 1K,L). Notably, the most efficient recovery in all mobility parameters was observed on day 5 of adulthood (Figure 1F,H,J,L). In contrast, nematodes fed with 10% of *sca-1* RNAi-expressing bacteria did not show any impact on the mobility parameters measured (Figure 1F,H,J,L). Notably, *sca-1* silencing in wild-type nematodes (Appendix A) also improves mobility parameters, although the major effect is observed in young adults (Appendix A). This observation points out that mobility improvement driven by *sca-1* knock-down does not rely on Aß_(1-42)_-overexpression. Collectively, these results indicate that robust *sca-1* silencing prevent Aß_(1-42)_-induced paralysis and Aß_(1-42)_-induced decline in locomotion parameters, suggesting that functional inhibition of SERCA might provide a sustained benefit to mitigate Aß_(1-42)_-induced muscle function decline overtime.

Loss of ER Ca^2+^ buffering capacity induced by *crt-1* silencing has been shown to counteract Aß_(1-42)_-induced toxicity in body-wall muscle cells, improving worm mobility [27]. Given that *sca-1* silencing might also contribute to this phenotype by targeting luminal ER Ca^2+^ levels, we tested whether the effects of *sca-1* and *crt-1* silencing were additive. Simultaneous *sca-1*/*crt-1* silencing on Aß_(1-42)_-overexpressing worms also induced a significant reduction in paralysis at days 4 and 8 of adulthood (Appendix A). The percentage of non-paralyzed worms upon *sca-1*/*crt-1* silencing almost doubled compared with single gene silencing on day 4, but was reduced by half on day 8 of adulthood (Appendix A). In free mobility assays *sca-1*/*crt-1* silencing also prevented Aß_(1-42)_-driven reduction in the mobility parameters at the level of length and maximum speed (Appendix A) without impacting on parameters of mean speed (Appendix A). However, double *sca-1*/*crt-1* silencing did not outperform on free mobility parameters compared to the effect of single gene silencing (Figure 1E–L). These results suggest that *sca-1* and *crt-1* silencing might counteract Aß_(1-42)_-induced toxicity in body-wall muscle by targeting the same cellular process.

### 2.2. Sca-1 Silencing Prevents the Development of Early Markers of Aβ_(1-42)_-Driven Toxicity

Even though Aβ_(1-42)_ is not expressed in pharyngeal muscle, disrupted pharyngeal pumping is an earlier marker of Aβ_(1-42)_-driven toxicity in worms overexpressing Aβ_(1-42)_ in body-wall muscles (Figure 2A) [28]. To assess the impact of *sca-1* on this earlier marker of Aβ-induced toxicity, we compared pumping rates across the first five days of adult life. As shown in Figure 2B, wild-type worms reach a maximum in pumping rates at day 2 to progressively decline. As anticipated, Aß_(1-42)_-overexpression significantly decreased pumping rates over the first 5 days of adult life. Consistent with a preventive role of *sca-1* silencing on Aβ_(1-42)_-driven toxicity, *sca-1* RNAi fully restored the pumping rates of Aβ_(1-42)_-overexpressing worms to the levels of the wild type (Figure 2B,C).

The mechanisms mediating this cell non-autonomous regulation are largely unknown (Figure 2A). Given that mitochondrial dysfunction is a key contributor to Aβ-induced toxicity [29,30] and the high energy requirements of the pharynx muscle [31], we hypothesized that Aß_(1-42)_-driven body-wall muscle dysfunction may mediate its impact on pharynx pumping rates by impairing pharynx mitochondrial bioenergetics. Accordingly, Aß_(1-42)_-overexpressing nematodes exposed to a temperature challenge produce larger amounts of mitochondrial superoxide ions in pharynx than wild-type worms at day 4 and day 8 of adulthood (Figure 2D,E). Robust *sca-1* silencing (100% RNAi) in Aß_(1-42)_-overexpressing worms completely reversed Aß_(1-42)_-induced superoxide production to the wild-type levels on both days measured (Figure 2D,E), whereas mild *sca-1* silencing (10% RNAi) did not have any measurable influence (Figure 2D,E). Interestingly, *sca-1* silencing also reduced mitochondrial superoxide production in N2 worms growing at the same temperature that triggers the paralysis phenotype in the GMC101 strain (25 °C), but the effect was lost at regular growing temperature (20 °C) (Figure 2F). At the whole-body level, oxygen consumption rates (OCR) were significantly increased under basal (Figure 2G,H) and uncoupling conditions (Figure 2G–I) in Aß_(1-42)_-overexpressing nematodes. These results suggest that Aß_(1-42)_ accumulation exacerbates respiratory chain activity, enhancing electron leak and superoxide production. *sca-1* silencing in Aß_(1-42)_-overexpressing nematodes did not impact on basal respiration, although it compromised the spare respiratory capacity (Figure 2G,I). Pharynx mitochondrial membrane potential was unaltered in Aß_(1-42)_-overexpressing nematodes, and *sca-1* RNAi did not induce further changes (Figure 2J,K). These results indicate that the electrical component of the proton motive force for ATP synthesis is not affected neither by Aß_(1-42)_-overexpression nor *sca-1* silencing, and suggest that Aß_(1-42)_-induced superoxide production is not driven by changes in the mitochondrial membrane potential.

### 2.3. Sca-1 Silencing Restores the Mitochondrial Network Organization in the Body-Wall Muscle

The spatial organization of mitochondrial networks in body-wall muscle cells displays a characteristic distribution, forming parallel rows of moderately elongated mitochondrial units whose spatial disposition is conditioned by sarcomers [32]. Loss of sarcomere and mitochondrial organization in body-wall muscle cells is associated with muscle dysfunction [33]. We previously reported that Aß_(1-42)_-overexpression disrupts body-wall mitochondrial shape and networks [27]. Here, we visualized body-wall mitochondrial networks with TMRE to test whether *sca-1* impacts mitochondrial shape andnetwork organization in Aß_(1-42)_-overexpressing worms (Figure 3A). Image analysis revealed that *sca-1* silencing modifies neither the cell area occupied by mitochondria, nor the number of mitochondrial units per cell nor their shapes, measured as aspect ratio (Figure 3B). We also performed a mitochondrial linearity assessment to evaluate mitochondrial spatial disposition along the longitudinal axis of the body-wall muscle cells. Fluorescence line profile data on mitochondrial masks (Figure 3C,D) revealed that in Aß_(1-42)_-overexpressing worms, mitochondrial longitudinal axis failed to align along a uniform linear path, resulting in increased discontinuities along the muscle fibers (Figure 3D,E). Notably, this disrupted mitochondrial arrangement was partially rescued by *sca-1* silencing, which led to increased mitochondrial coverage along the longitudinal axis and a significant increase in linearity values (Figure 3D,E). The fluorescence intensity of body-wall muscle cells in TMRE-loaded worms was also quantified as a readout of the mitochondrial membrane potential (ΔΨ_m_). No significant differences in ΔΨ_m_ were observed across conditions (Figure 3F), indicating that neither Aß_(1-42)_ expression nor *sca-1* silencing compromised the electrical component of the proton motive force for ATP-synthesis nor the driving force for the transport of proteins and metabolites across the inner mitochondrial membrane relying on physiological ΔΨ_m_. We conclude that although *sca-1* silencing does not have a significant impact on body-wall muscle mitochondrial bioenergetics, it partially restores the mitochondrial network organization in the body-wall muscle of Aß_(1-42)_-overexpressing worms.

### 2.4. Sca-1 Knockdown Reduces Aß_(1-42)_ Aggregation and Oligomerization in Body Wall Muscle Cells

To explore the protective mechanisms preventing Aß_(1-42)_-induced muscle dysfunction triggered by *sca-1* silencing, we tested whether improved muscle performance in *sca-1* knockdown worms correlated with changes in the number and size of Aß_(1-42)_ aggregates. In vivo aggregates were visualized and quantified using the amyloidogenic dye Thioflavin T (ThT). As previously reported, GMC101 Aß_(1-42)_-overexpressing worms produce significant levels of human Aß_(1-42)_ mRNA (Figure 4C) and show accumulation of Aß_(1-42)_ aggregates within the head area of the body-wall muscle (Figure 4A). Nematodes exposed to *sca-1* RNAi from hatching significantly reduced the number of visible aggregates (Figure 4A). This is reflected in a reduction in the percentage of the Thioflavin T-stained area relative to the total area of about 32% compared to control animals (Figure 4B). To investigate whether this reduction was associated with changes in the overall levels of Aß_(1-42)_ peptide and its oligomerization status, we performed western-blot analysis on total protein extracts using Aß_(1-42)_ antibodies. Samples were resolved under denaturing conditions, allowing the separation of monomeric (~4 kDa), dimeric (~8 kDa), and higher-order oligomeric forms of Aß (Figure 4D). Densitometric analysis revealed that monomers and dimers levels were comparable between control and *sca-1* RNAi conditions (Figure 4E,F). However, the abundance of trimeric/tetrameric and higher-order oligomers (>12 kDa) was consistently decreased upon *sca-1* silencing (Figure 4G,H). Indeed, the total Aß_(1-42)_ levels were significantly reduced in *sca-1* knockdown worms (Figure 4I). These results cannot be attributable to reduced Aß_(1-42)_ mRNA production, as mRNA levels were unaffected by *sca-1* silencing (Figure 4J). The results suggest that inhibition of Ca^2+^ pumping into the ER via *sca-1* diminishes Aß oligomerization and aggregation. This mechanism might explain the reduced sensitivity of *sca-1* knockdown worms to Aß_(1-42)_ toxicity in body-wall muscle cells.

### 2.5. Sca-1 Silencing Partially Restores Aß_(1-42)_-Induced Remodeling of Cytosolic Ca^2+^ Dynamics in Muscle Cells

Accumulation of Aß_(1-42)_ disrupts cellular Ca^2+^ homeostasis in both cellular and murine models [6]. Therefore, we decided to measure the impact of Aß_(1-42)_ overexpression on cytosolic Ca^2+^ dynamics and test whether *sca-1* loss of function can restore any potential disruption caused by Aß_(1-42)_. To measure [Ca^2+^]_cyt_, we used transgenic strains expressing the FRET-based Ca^2+^-sensitive indicator YC2.1 in the cytosol of body-wall muscle cells, including vulva muscles. Spontaneous Ca^2+^ oscillations in vulva muscle cells were monitored for 15 min. Figure 5A shows representative 200 s [Ca^2+^]_cyt_ records of control worms, Aß_(1-42)_-overexpressing worms, and the impact of *sca-1* silencing on Aß_(1-42)_-overexpressing worms. The figure displays the fluorescence ratio F535/F480, along with individual F535 and F480 fluorescence data. Both wavelengths show mirror-image changes, reflecting true fluctuations in [Ca^2+^]_cyt_. Aß_(1-42)_ overexpression increased the peak width at half-height (Figure 5B) without affecting the peak height (Figure 5C) suggesting slower kinetics of spontaneous cytosolic Ca^2+^ transients, a phenomenon that might contribute to Aß_(1-42)_ toxicity. This slower kinetics of the cytosolic Ca^2+^ transients was properly revealed by reduced rise and decay cytosolic Ca^2+^ rates (Figure 5D,E). *sca-1* silencing in Aß_(1-42)_ overexpressing worms partially restored the kinetic parameters analyzed. *sca-1* knockdown reduced the peak width (Figure 5B) and enhanced the peak height (Figure 5C). Interestingly, only decay rates were significantly rescued by *sca-1* knockdown (Figure 5D,E). Basal cytosolic Ca^2+^ levels inferred from FRET ratios on vulva muscles were not affected by Aβ_(1-42)_ overexpression and *sca-1* silencing (Appendix A). These results suggest that the reduced sensitivity of *sca-1* knockdown worms to Aß_(1-42)_ toxicity in body-wall muscle cells might be partially mediated by remodeling or restoration of the cytosolic Ca^2+^ dynamics.

## 3. Discussion

Disruptions in intracellular calcium (Ca^2+^) homeostasis have been consistently linked to the pathogenesis of Alzheimer’s disease (AD). Among the diverse Ca^2+^ transport systems regulating intracellular Ca^2+^ dynamics, the activity of the sarco-endoplasmic reticulum Ca^2+^-ATPase (SERCA) has been shown to modulate phenotypic outcomes in preclinical AD models, although the cellular mechanisms underlying its influence remain poorly defined. Here, we show that RNAi-mediated silencing of *sca-1*, which encodes the *C. elegans* ortholog of SERCA, partially ameliorates Aβ_(1-42)_-induced phenotypes, including paralysis, impaired locomotion, and reduced pharyngeal pumping (Figure 1). These findings suggest that partial inhibition of *sca-1*, and thus a reduced capacity for ER Ca^2+^ sequestration, confers protection against Aβ_(1-42)_-mediated toxicity. Furthermore, overexpression of Aβ_(1-42)_ in body-wall muscle cells was found to perturb cytosolic Ca^2+^ transient dynamics without significantly affecting basal cytosolic Ca^2+^ levels. Notably, *sca-1* silencing partially restored the dynamics of cytosolic Ca^2+^ transients (Figure 5), suggesting that its protective effect may stem from both compensatory responses of alternative Ca^2+^ transport systems and a reduction in the ER Ca^2+^ level. This interpretation is further substantiated by our previous findings that *crt-1* silencing, which disrupts the ER major Ca^2+^ buffering mechanism, also prevents Aß_(1-42)_-induced body-wall muscle dysfunction [27], and the lack of synergistic effect on mobility of upon double *sca-1*/*crt-1* silencing (Appendix A). Collectively, these results highlight the pathological role of excessive ER Ca^2+^ accumulation in Aβ toxicity and suggest that targeting ER Ca^2+^ homeostasis may represent a potential therapeutic strategy in AD.

To elucidate the mechanisms underlying the protective effect of *sca-1* on muscle function, we demonstrate that *sca-1* silencing is associated with reduced levels of Aβ_(1-42)_ aggregates, enhanced mitochondrial network integrity in body-wall muscle cells, and decreased Aβ_(1-42)_-induced superoxide production in pharyngeal mitochondria. In regard to mitochondrial function, it is well established that sustained elevations in mitochondrial matrix Ca^2+^, especially when combined with other stressors such as increased reactive oxygen species (ROS), can trigger the opening of the mitochondrial permeability transition pore (mPTP) [34]. Activation of the mPTP disrupts the inner mitochondrial membrane ion, metabolite and electrical gradients, leading to mitochondrial dysfunction and the release of pro-apoptotic factors [34]. Our findings indicate that Aβ_(1-42)_ overexpression induces mitochondrial stress, as evidenced by elevated superoxide levels and increased basal oxygen consumption (Figure 2). Under these conditions, mitochondria become more susceptible to mPTP opening in response to Ca^2+^ overload. We hypothesize that *sca-1* silencing, which leads to reduced ER Ca^2+^ levels, diminishes ER-to-mitochondria Ca^2+^ transfer. As a result, mitochondria are less prone to mPTP opening, thereby maintaining functional competence under stress conditions. Moreover, we have previously shown that submaximal *sca-1* silencing in post-developed *C. elegans* promotes lifespan and health span [20]. Our findings suggest that the pro-survival effect is mediated by a reduced ER-to-mitochondria Ca^2+^ transfer and requires activation of AMP-activated protein kinase (AMPK) alongside inhibition of the mechanistic target of rapamycin (mTOR) pathway [21]. Based on this evidence, it is plausible to hypothesize that a metabolic shift favoring catabolic pathways while repressing anabolic activity may compensate for Aβ_(1-42)_-induced mitochondrial dysfunction, thereby contributing to the protective phenotype observed in our model.

Our results demonstrate that overexpression of Aβ_(1-42)_ in body-wall muscles significantly attenuates the kinetics of spontaneous cytosolic Ca^2+^ transients in muscle cells, a phenotype that is partially rescued by *sca-1* knockdown (Figure 5). Precise spatial and temporal regulation of cytosolic Ca^2+^ signaling is critical for muscle function [35]. Slower Ca^2+^ transient kinetics may impair the contraction-relaxation cycles of muscle and promote Ca^2+^-dependent cytotoxicity, reminiscent of excitotoxic mechanisms [36]. *sca-1* knockdown was found to partially normalize the temporal characteristics of Ca^2+^ transients, suggesting a modulatory role in mitigating Aβ_(1-42)_-induced Ca^2+^ dysregulation. Specifically, *sca-1* depletion enhanced the decay kinetics of cytosolic Ca^2+^ signals (Figure 5E). Given the anticipated reduction in *sca-1*-mediated ER Ca^2+^ reuptake under knockdown conditions, we hypothesize that reduced *sca-1* activity induces compensatory remodeling of Ca^2+^ fluxes across cellular membranes. This may involve upregulation of alternative clearance pathways, such as plasma membrane Ca^2+^-ATPases (PMCAs) or Na^+^/Ca^2+^ exchangers (NCXs), thereby facilitating accelerated Ca^2+^ extrusion and partially restoring cytosolic Ca^2+^ homeostasis.

Another important finding of this manuscript is the reduction in Aβ_(1-42)_ aggregates and high molecular weight Aβ_(1-42)_ oligomers upon *sca-1* silencing. We noticed that our new results contrast with those previously reported, indicating *sca-1* silencing did not impact on the levels of Aβ_(1-42)_ in the same *C. elegans* strain we have used here [27]. The discrepancy may be explained by the use of an RNAi dose 10 times lower in those experiments. Given the established toxicity of oligomeric Aß species, reduced Aβ_(1-42)_ accumulation might therefore explain *per se* the improvement in the parameters of body-wall muscle functionality observed in GMC101 *sca-1* knockdown worms (Figure 1). In our experimental setting, Aβ_(1-42)_ steady state levels result from a balance between transcriptional overproduction of Aβ_(1-42)_ driven by the muscle-specific promoter *unc-54*/*myo-4* and the action of Aβ_(1-42)_ clearance mechanisms. Since *sca-1* silencing did not impact on Aβ_(1-42)_ mRNA levels, it can be hypothesized that lack of *sca-1* may reduce Aβ_(1-42)_-deposits by activating protein clearance mechanisms. Mechanisms previously associated with Aβ-clearance include ubiquitin-proteosome degradation, chaperone-mediated autophagy, or macro-autophagy [37]. A mechanistic interpretation of this phenomenon may be hypothesized. Partial *sca-1* loss-of-function elicits endoplasmic reticulum (ER) stress, thereby activating the unfolded protein response (UPR) and inducing transcriptional upregulation of chaperones, including *hsp-4* in *C. elegans* [38,39,40]. The UPR signaling cascade facilitates the induction of macroautophagy, which serves as a proteostatic mechanism for the degradation of misfolded or aggregated proteins [41]. Importantly, amyloid-β (Aβ) has been demonstrated to be a substrate for autophagy-mediated lysosomal degradation in both rodent models and *C. elegans* [41,42]. Collectively, we propose that *sca-1* RNAi-driven UPR activation, coupled with enhanced chaperone expression and autophagic flux, may potentiate the catabolism of Aβ species, thereby promoting the clearance of proteotoxic aggregates and the re-establishment of intracellular proteostasis.

Taken together, current evidence from *C. elegans* supports the notion that a partial reduction in endoplasmic reticulum (ER) Ca^2+^ levels may activate pro-survival signaling pathways that enhance longevity and confer protection against Aβ_(1-42)_-induced toxicity. Direct in vivo measurements of ER Ca^2+^ concentrations following *sca-1* silencing would provide critical validation of this hypothesis. Notably, these findings in *C. elegans* stand in contrast to observations in mouse models, where pharmacological activation of SERCA has been reported to attenuate Alzheimer’s disease (AD) pathology [14,15,16]. This apparent divergence underscores the complexity of ER Ca^2+^ dynamics in neurodegeneration and highlights the importance of model-specific context in interpreting therapeutic strategies. Transgenic *C. elegans* models that ectopically overexpress Aβ in body-wall muscle cells constitute a robust system for investigating mechanisms of protein aggregation and proteotoxic stress. However, their translational relevance to Alzheimer’s disease remains limited, as they do not reproduce fundamental aspects of the disorder’s neuropathophysiology, including synaptic dysfunction, dysregulated neurotransmission, and progressive neuronal degeneration. Interestingly, brain-specific partial loss of SERCA2 (ATP2A2) function has been reported to cause only mild behavioral abnormalities [43]. However, its potential protective role against pathology in mouse models of Alzheimer’s disease remains unexplored.

## 4. Materials and Methods

### 4.1. C. elegans Strains and Maintenance

*Caenorhabditis elegans* strains were cultured at 20 °C and maintained on Nematode Growth Medium (NGM) plates seeded with live *Escherichia coli* (OP50) unless otherwise specified. Strains used included wild-type, N2, and transgenic strain GMC101 (dvIs100 [unc-54p::A-beta-1-42::unc-54 3′-UTR + mtl-2p::GFP]), were provided by the *C. elegans* Genetics Center (CGC, University of Minnesota, MN, USA), which is funded by NIH Office of Research Infrastructure Programs (P40 OD010440). The following *C. elegans* strain was generated for this study CAG001 (*smg-1*(*cc546ts*); dvIs27 X; pmyo3::YC2.1) by crossing CL4176 (*smg-1*(*cc546*) *I*; dvIs27 X) with AQ2121 (*pmyo3::YC2.1*) kindly provided by Drs. Robyn Branicky and W. Schafer, MRC Laboratory of Molecular Biology, Cambridge, United Kingdom. The AQ2121 strain expresses the ratiometric Ca^2+^ sensor YC2.1 in body wall muscle cells cytosol. Progeny was recrossed to obtain integrated *pmyo3::YC2.1* in homozygosis and then genotyped by PCR to confirm the presence of ß-amyloid peptide.

### 4.2. RNAi Feeding Worms and Experimental Specifications

The HT115 RNAi clone for *sca-1* (K11D9.2, Ahringer library) and the empty vector (L4440) were kindly provided by Dr. Malene Hansen from the Sanford Burnham Prebys Medical Discovery Institute in La Jolla, CA, United States. RNAi was induced by feeding the worms with transgenic bacteria seeded on solid NGM plates, which contained 1 mM isopropyl-β-D-thiogalactoside (IPTG), 50 μM carbenicillin, and 15 μM fluorodeoxyuridine (FUdR), when necessary. The empty vector (HT115 or L4440) and the *sca-1* (K11D9.2) RNAi clone were cultured overnight at 37 °C in LB medium with 125 μM ampicillin, as described in [21]. Once they reached a DO_595_ = 0.6, *sca-1* RNAi and empty vector L4440 bacteria were mixed to the desired percentage and seeded onto NGM plates (150 μL on 35 mm plates or 400 μL on 55 mm agar plates). The plates were then incubated for 3 days at 20 °C. Finally, synchronized eggs were placed on RNAi-seeded plates without FUdR to ensure effective knockdown of *sca-1* and proper development. Worms were maintained at 20 °C until amyloid aggregation and paralysis phenotype induction.

### 4.3. RNA Extraction

500–600 worms at day 3 of adulthood were collected and washed three times with M9 solution. The samples then underwent three freeze/thaw cycles in liquid nitrogen and were subsequently stored at −80 °C. For homogenization, the frozen samples were grounded using Bel-Art^®^ Disposable Pestles (BAF199230000), and RNA was extracted using a phenol-chloroform method as previously described in [44]. The quality and concentration of the RNA were assessed using a NanoDrop One spectrophotometer (Thermo Fisher, Waltham, MA, USA) and 1% agarose gel electrophoresis to ensure the integrity of the extracted RNA.

### 4.4. cDNA Synthesis and qRT-PCR

RNA samples were incubated 30 min at RT with DNase (RNase-Free DNase Set, 79254, Quiagen, Barcelona, Spain) to eliminate any residual genomic DNA contamination. 1 μg of RNA was used for reverse transcription using the iScript cDNA synthesis kit (BioRad, Madrid, Spain) according to manufacturer instructions, using random primers. After cDNA synthesis, reactions were performed using the SYBR Green Master Mix (Applied Biosystems/Thermo Fisher Scientific, Waltham, MA, USA) on a LightCycler 480 PCR system (Roche Applied Science, Barcelona, Spain). Primers used were: *sca-1* forward, CTTCCAGCCACTGCTCTCGGATTC; *sca-1* reverse, CTGGTAGTAGGTGATCTGTGGTCC; *hAß* forward, GCAGAATTCCGACATGACTCAG; *hAß* reverse, GCCCACCATGAGTCCAATGA; *cdc-42* forward, CTGCTGGACAGGAAGATTACG; *cdc-42* reverse, CTCGGACATTCTCGAATGAAG. *cdc-42* was used as the endogenous control gene for quantification. The following program parameters were used for all amplifications: 95 °C for 10 min, followed by 45 cycles at 95 °C for 15 s, 60 °C for 30 s and 72 °C for 30 s, and finally one cycle at 95 °C for 20 min, 65 °C for 1 min, and 97 °C for 5 min. Assays were performed using three biological replicates, each consisting of three technical triplicates.

### 4.5. Paralysis Assays

Bleaching-synchronized GMC101 and N2 populations were maintained at 20 °C until they reached young adult stage. Then, 300 worms per experimental condition were transferred to fresh NGM agar plates containing *sca-1* RNAi, L4440, or a mixture of both at different ratios. Worms were then shifted from 20 °C to 25 °C and paralysis was assessed at days 4 and 8 after the temperature shift. Worms were scored as paralyzed if they failed to complete full-body movement, either spontaneously or upon touch stimulation, following the criteria previously described in [26]. Assays were performed using three biological replicates, each consisting of 100–150 worms per condition.

### 4.6. Tracking Assays

Tracking assays were performed on days 1, 3, and 5 of adulthood, as previously described in [27]. Briefly, 10–15 young adult worms were placed on 35 mm NGM plates containing *sca-1* RNAi, L4440, or a mixture of both, and shifted from 20 °C to 25 °C. Thirty-second video were recorded under a stereomicroscope (Leica S9i, Leica Microsystems Srl, Milan, Italy) and analyzed with the Fiji plugin “wrMTrck”.

### 4.7. Superoxide Measurements

Mitochondrial superoxide levels were measured alongside paralysis assays, using the same plates for both tests. For each experimental condition, 10–20 worms were collected in 250 µL of M9 buffer with an eyelash to prevent bacterial contamination. MitoSOX^TM^ was dissolved in M9 buffer to a final concentration of 2 µM in a total volume of 500 µL per condition. The nematodes were incubated with this solution for 4.5 h in the dark under shaking conditions and then washed with M9 medium. Worms were then placed on an empty plate and carefully transferred onto 2% agarose pads with a drop of 20 mM tetramisole for immobilization. Images of the pharynx region were captured using a Nikon ECLIPSE Ni-E microscope (Thermo Fisher, Waltham, MA, USA) equipped with a TRITC filter (excitation 540/25, emission 605/55) and a DS-Ri2 camera. Fluorescence intensity was quantified using ImageJ 1.54g software. Each biological replicate was normalized to the mean fluorescence intensity of the N2 strain. This procedure was repeated nearly three times.

### 4.8. Mitochondrial Membrane Potential Measurements

Mitochondrial membrane potential was measured on day 4 of adulthood in N2 and GMC101 worms. For each experimental condition, 10–20 worms were collected in 250 µL of M9 buffer using an eyelash to avoid bacterial contamination. TMRE was dissolved in M9 buffer to a final concentration of 0.1 µM in a total volume of 500 µL per condition. An additional negative control tube was prepared by adding 100 µM of the protonophore CCCP. The nematodes were incubated with this solution for 3.5 h in the dark with shaking, followed by washing with M9 buffer. Worms were then placed on an empty plate and carefully transferred to 2% agarose pads containing a drop of 20 mM tetramisole for immobilization. Images of the pharynx region were captured using a Nikon ECLIPSE Ni-E fluorescence microscope equipped with a TRITC filter (excitation 540/25, emission 605/55) and a DS-Ri2 camera (Thermo Fisher, Waltham, MA, USA), or a Leica TCS SP5 confocal microscope (Leica Microsystems Srl, Milan, Italy) (excitation 549 nm, emission 574 nm). Each biological replicate was normalized to the mean fluorescence intensity of the N2 strain.

### 4.9. Body-Wall Muscle Mitochondrial Organization

Body-wall muscle mitochondrial organization was measured as previously described in [29]. Day 4 adult worms were incubated for 24 h on NGM plates containing 5 μM TMRE added before pouring. Worms were then washed with M9 and transferred for 30 min to standard NGM plates in the dark. Thereafter, worms were immobilized on 2% agarose pads with 20 mM tetramisole and imaged using a Leica TCS SP5 confocal microscope (excitation 549 nm, emission 550–612 nm). Images were analyzed using the ImageJ 1.54g plugin ‘Mitochondria Analyzer’ [45] to calculate the morphological parameters: aspect ratio, mitochondrial number per cell and total mitochondrial area. To assess the linearity of mitochondria along *C. elegans* body-wall muscle cells, the same set of confocal microscopy images were re-analyzed. Mitochondria were first segmented using the ImageJ 1.54g plugin Mitochondrial Analyzer. Following segmentation, linear mitochondrial organization was evaluated using ImageJ 1.54g. For each image, three independent 200-pixel-length lines were drawn randomly across the mitochondrial mask in an unbiased manner. The intensity profile along each line was extracted and converted to binary data: intensity values of zero were assigned a value of 0, and all non-zero intensity values were assigned a value of 1. This process generated 200 binary data points per line. The binary values were then summed, yielding a score between 0 and 200. A score of 200 indicated uninterrupted mitochondrial presence along the entire line, whereas lower values reflected increasing discontinuity in the mitochondrial arrangement.

### 4.10. Oxygen-Consumption Assays

Mitochondrial respiration was assessed using the XFe24 extracellular flux analyzer (Agilent Technologies, Barcelona, Spain), following the protocol established by [46] with minor adaptations. Synchronized day 4 adult worms were collected in 1 mL of M9 buffer with an eyelash to minimize bacterial contamination and placed briefly on an unseeded NGM plate. Groups of 10–15 worms were transferred into each well of a Seahorse XF24 cell culture microplate containing 500 μL of M9 buffer. Oxygen consumption was recorded every 5 min using a cycle of 2 min mixing, 30 s waiting, and 2 min measuring at 25 °C. When indicated, the following modulators of mitochondrial respiration were added: FCCP (10 μM) and sodium azide (50 mM). The data were normalized to worm number using the Seahorse Analytics software version 2.6 and exported to GraphPad Prism 9.4 for analysis. Each condition was tested in three independent biological replicates, each comprising four technical replicates.

### 4.11. Calcium Measurements

Body-wall muscle cytosolic [Ca^2+^] ([Ca^2+^]_cyt_) measurements were performed using the strains AQ2121 and CAG001. Both express the ratiometric Ca^2+^ sensor YC2.1 in body-wall muscle cells cytosol. Briefly, worms were glued (Dermabond Topical Skin Adhesive, Johnson and Johnson, Madrid, Spain) on an agar pad (2% agar in M9 buffer) and the coverslip containing the glued worm was mounted in a chamber in the stage of a Zeiss Axiovert 200 inverted microscope. Fluorescence was excited at 430 nm using a Cairn monochromator (4 nm bandwidth, continuous excitation). Emitted fluorescence was collected with a Zeiss C-apochromat 40 × 1.2 W objective and filtered using a 450 nm long pass dichroic mirror and a Cairn Optosplit II emission image splitter to obtain separate images at 480 and 535 nm emission. The splitter contained emission filters DC/ET480/40 and DC/ET535/30, and the dichroic mirror FF509-FDi01–25 × 36 (all from Chroma Technology, Olching, Germany). Simultaneous 200 ms images at the two emission wavelengths were recorded continuously (2.5 Hz image rate) by a Hamamatsu ORCA-ER camera, to obtain 535/480 nm fluorescence ratio images values of a region of interest enclosing the vulva area. Experiments were performed during 15 min of continuous recording. Fluorescence was recorded and analyzed using the Metafluor program (Universal Imaging). Then, the fluorescences obtained at 480 and 535 nm emission were analyzed off-line using a home-made specific algorithm designed to first identify the Ca^2+^ peaks by looking for mirror changes in both wavelengths and then to calculate for each Ca^2+^ peak in each experiment the total width and the width at mid-height expressed in seconds, the height obtained as a percentage of ratio change from the base to the peak, the time to rise from the base to the peak and the time to return to the base from the peak.

### 4.12. Aß In Vivo Aggregate Staining

Aß aggregates were measured in vivo using Thioflavin-T (ThT). The fluorescent probe ThT was dissolved in M9 buffer and added on NGM plates to a final concentration of 50 µM. 24 h before imaging, worms were transferred to RNAi seeded plates containing ThT. After ThT incubation period, worms were washed three times with M9 buffer and transferred to standard NGM plates (without ThT) for 30 min in the dark to reduce background fluorescence. Worms were then mounted on 2% agarose pads containing 20 mM tetramisole for immobilization and covered with a coverslip. Imaging was performed using a Leica TCS SP5 confocal microscope. ThT fluorescence was excited with an argon laser at 458 nm, and emission was collected between 470 and 553 nm. Z-stacks were acquired at 0.3–0.5 µm intervals to capture the entire height of the worm. Image analysis was performed using ImageJ 1.54g. A consistent threshold was applied across images to detect visible aggregates, and regions of interest (ROIs) were semi-automated identified using the Particle Analyzer plugin. ROIs were manually reviewed. The area of each ROI was measured, and the sum of all ROIs in a worm was calculated. This total aggregate area was then normalized to the total area of the worm to obtain the fraction of worm area occupied by Aβ aggregates.

### 4.13. Protein Extraction and Wester Blotting

For protein extraction, approximately 400–500 day-4 adult worms were collected and washed with M9 buffer. The pellet was resuspended in 100 μL of RIPA lysis buffer supplemented with protease inhibitors (Roche) and PhosSTOP phosphatase inhibitor cocktail (Roche), and subjected to three cycles of sonication using a Vibra-Cell 75,115 sonicator (Bioblock Scientific, Barcelona, Spain) at 4 °C. Each cycle consisted of three 10 s pulses (10 s ON/10 s OFF) at 39% amplitude. Following lysis, samples were centrifuged at 14,000× *g* for 20 min at 4 °C to remove debris. The supernatant was collected, and total protein concentration was quantified using the Pierce^®^ BCA Protein Assay Kit (ThermoFisher, Madrid, Spain). Equal amounts of protein (22 μg) were resolved by SDS-PAGE (Bio-Rad, Madrid, Spain) and transferred onto nitrocellulose membranes using the i-blot (Bio-Rad, Madrid, Spain). Membranes were blocked and incubated with primary antibodies against β-amyloid (6E10, BioLegend, Barcelona, Spain; 1:3000) and ß-actin (A3854, Merck, Barcelona, Spain; 1:200,000). After washing, membranes were incubated with HRP-conjugated secondary antibodies (P0447, DakoCytomation, Glostrup, Denmark; 1:10,000), and protein bands were visualized by enhanced chemiluminescence (SuperSignal™ West Pic PLUS Chemiluminescent Substrate, ThermoFisher, Madrid, Spain).

### 4.14. Statistical Analysis

Statistical analyses were performed using Prism software (GraphPad 9.4). All data sets were tested for normal distribution using the Kolmogorov–Smirnov test. Unless otherwise indicated, significance between two experimental groups was determined using a two-tailed unpaired Student’s *t*-test whereas group sets were analyzed using a one-way ANOVA (post hoc Tukey test). NS, not significant; * *p* < 0.05; ** *p* < 0.01; *** *p* < 0.001; **** *p* < 0.0001.

## 5. Conclusions

We conclude that partial loss of *sca-1* function enhances muscle performance in a *C. elegans* model of proteotoxicity characterized by Aβ_(1-42)_ accumulation in body-wall muscle cells. This improvement is associated with reduced levels of Aβ_(1-42)_ oligomers and aggregate formation, enhanced mitochondrial function, and the restoration of cytosolic Ca^2+^ transient kinetics. These observations support a model in which *sca-1* knockdown mitigates β-amyloid–induced toxicity by limiting Aβ_(1-42)_ oligomerization and promoting mitochondrial function through a mechanism that involves remodeling of cytosolic Ca^2+^ dynamics and partial ER Ca^2+^ depletion.

## Figures and Tables

**Figure 1 ijms-26-09126-f001:**
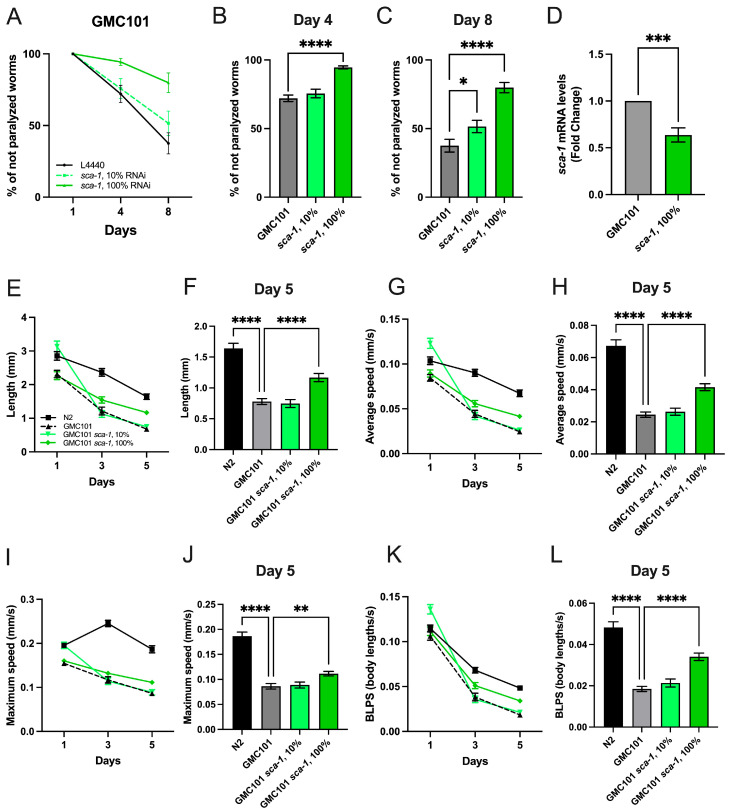
Impact of *sca-1* silencing on temperature-induced paralysis and mobility of GMC101 worms. (**A**) Percentage of non-paralyzed worms fed with control RNAi (L4440), *sca-1* RNAi, or a mixture of both, at day 1, 4, and 8 of adulthood. (**B**,**C**) Percentage of non-paralyzed worms at day 4 (**B**), and at day 8 (**C**). (**D**) Effect of *sca-1* RNAi on *sca-1* mRNA levels in GMC101 worms (n = 3, experiments). Panels (**E**,**G**,**I**,**K**) show the following mobility parameters of N2 and GMC101 fed with control RNAi (L4440), *sca-1* RNAi, or a mixture of both at day 1, 3, and 5 of adulthood: (**E**) length, (**G**) average speed, (**I**) maximum speed, (**K**) BLPS (body lengths per second). The following mobility parameters are represented at day 5 of adulthood: (**F**) length, (**H**) average speed, (**J**) maximum speed, and (**L**) BLPS (body lengths per second). For paralysis and mobility assays, results represent three biological replicates (n = 300 worms/condition and n @ 60 worms/condition, respectively). Data are mean ± s.e.m. * *p* < 0.05; ** *p* < 0.01; *** *p* < 0.001; **** *p* < 0.0001.

**Figure 2 ijms-26-09126-f002:**
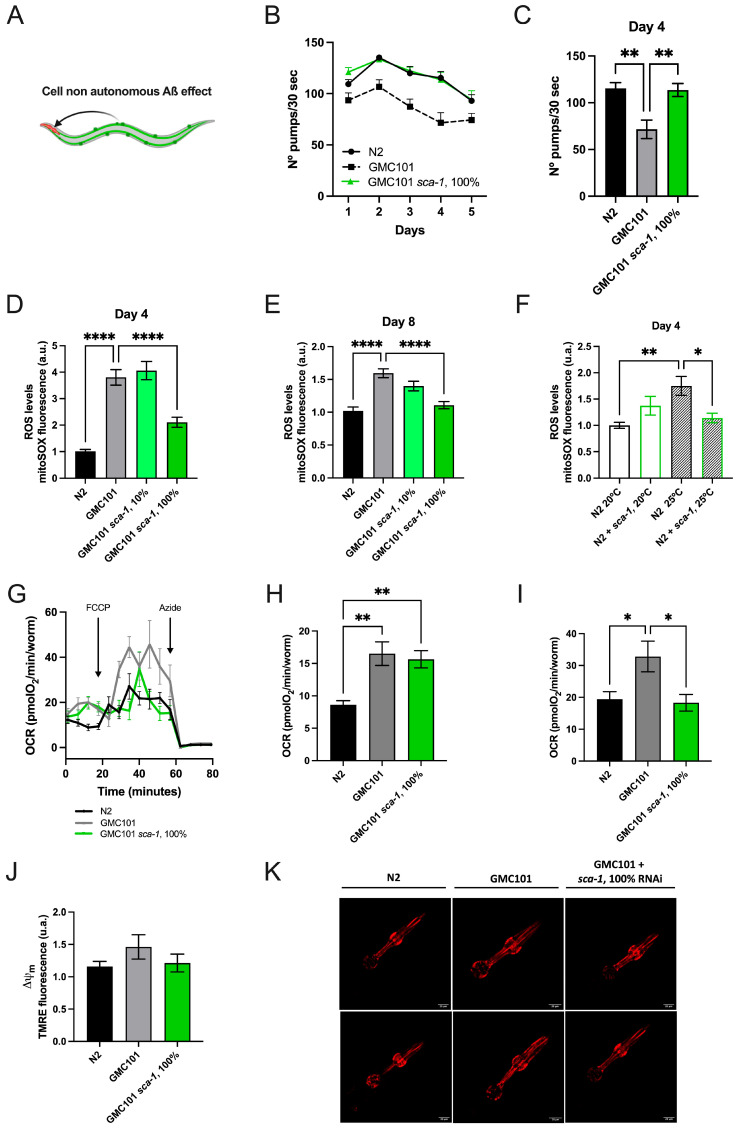
Impact of *sca-1* silencing on GMC101 mitochondrial function and mitochondrial enriched pharynx organ. (**A**) Representative scheme of pharynx cell non autonomous Aß effect. (**B**) Number of pumps/30 s of N2 and GMC101 fed with control RNAi (L4440) and 100% *sca-1* RNAi, at day 1, 2, 3, 4 and 5 of adulthood. (**C**) Number of pumps/30 s at day 4. (**D**) GMC101 mitochondrial superoxide production measurements with MitoSOX™ at day 4 of adulthood, and (**E**) at day 8 of adulthood. (**F**) N2 mitochondrial superoxide production measurements with MitoSOX™ at day 4 of adulthood, at 20 °C and 25 °C. (**G**) Oxygen consumption rates (OCR)., The protonophore FCCP, and the cytochrome C oxidase inhibitor azide, were added when indicated to obtain the maximum uncoupled respiration rate and nonmitochondrial respiration, respectively. (**H**) Effect of *sca-1* silencing on basal respiration. (**I**) Effect of *sca-1* silencing on maximal (uncoupled) respiration in the presence of FCCP. (**J**) Mitochondrial membrane potential measurements with TMRE at day 4 of adulthood. (**K**) Pharynx mitochondria membrane potential staining with TMRE at day 4 of adulthood. All experiments were completed with almost 3 biological replicates. Pumping experiments were performed using n = 15–20 worms/condition. Mitochondrial superoxide and mitochondrial membrane potential were performed using n = 50–80 animals/condition. Oxygen consumption measurements were performed using n = 15–20 worms/condition. Data are mean ± s.e.m. * *p* < 0.05; ** *p* < 0.01; **** *p* < 0.0001.

**Figure 3 ijms-26-09126-f003:**
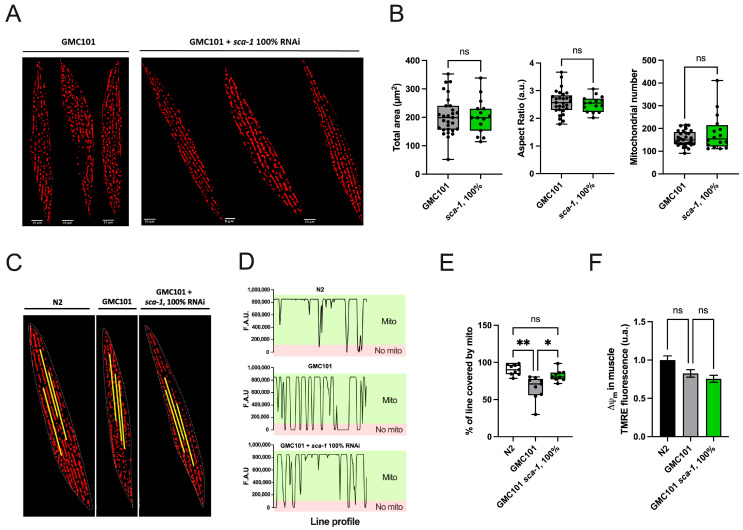
Impact of *sca-1* silencing on Aß-induced mitochondrial disorganization in body-wall muscles. (**A**) A series of representative confocal images of body wall muscle cells mitochondrial networks at day 4 of adulthood in GMC101 (overexpressing Aß) worms, and GMC101 worms upon *sca-1* RNAi silencing. (**B**) Mitochondrial morphology and organization parameters: total area covered by mitochondria/cell, aspect ratio, and number of mitochondria/cell. (**C**) Mitochondrial linearity across sarcomeric-muscle cells. (**D**) Representation of mitochondrial profile across a line of N2 and GMC101 fed with control RNAi (L4440) and 100% *sca-1* RNAi. The absence of mitochondria is scored at zero and highlighted in red. Values above zero indicate the presence of mitochondria and are highlighted in green. (**E**) Percentage of line covered by mitochondria. (**F**) Muscle mitochondrial membrane potential staining with TMRE at day 4 of adulthood. All experiments are the result of three biological replicates, each containing 3 technical replicates. Data are mean ± s.e.m. * *p* < 0.05; ** *p* < 0.01.

**Figure 4 ijms-26-09126-f004:**
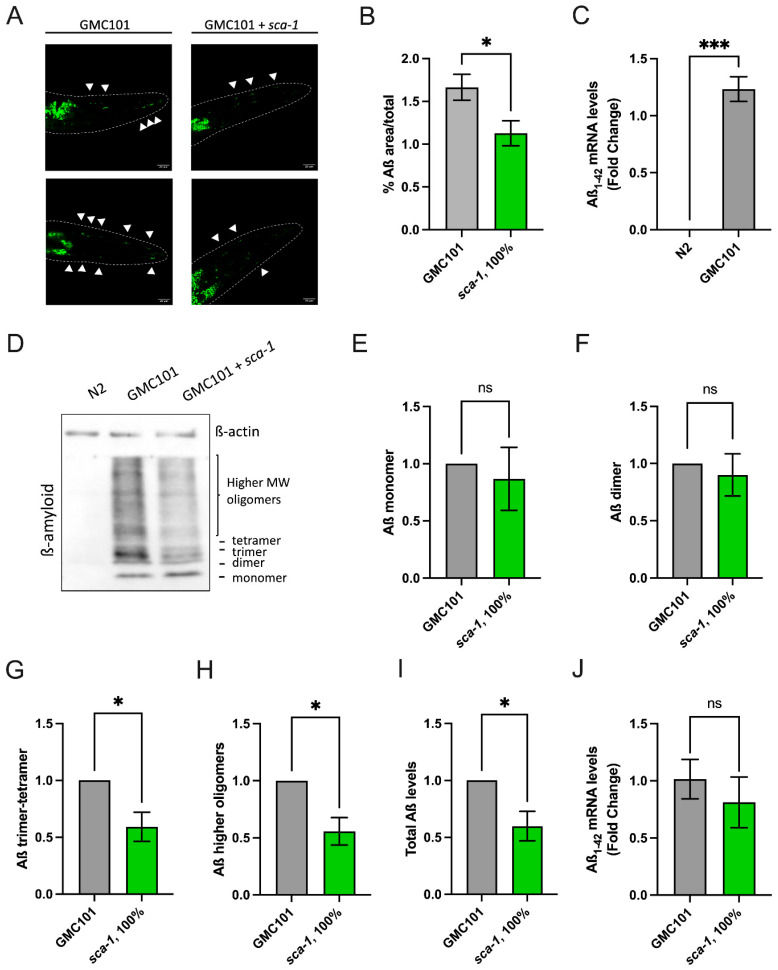
Impact of *sca-1* silencing on Aß aggregates. (**A**) Representative confocal images of Aß aggregates stained with Thioflavin-T located in muscle cells at the anterior region of the body wall muscle on day 4 nematodes. Arrows indicate Aß-aggregate localization. (**B**) Impact of *sca-1* silencing on the percentage of area occupied by Aß aggregates in the anterior region of the body wall muscle. (**C**) Aß mRNA levels in N2 versus GMC101 worms. (**D**) Aß levels and aggregation in GMC101 worms upon *sca-1* RNAi shown by Western blotting, a representative image of the three biological repeats performed. (**E**) Protein levels of Aß monomeric form. (**F**) Protein levels of Aß dimeric form. (**G**) Protein levels of Aß trimer-tetrameric form. (**H**) Protein levels of Aß higher molecular weight complexes. (**I**) Protein levels of total Aß peptide regardless of its aggregation degree. (**J**) Aß mRNA levels in GMC101 control (L4440) versus GMC101 100% *sca-1* RNAi worms. All experiments are the result of three biological replicates, each containing almost 3 technical replicates. ThT Aß-staining experiments were completed with n = 10–20 animals/experimental condition. Data are mean ± s.e.m. ns: *p* > 0.05; * *p* < 0.05; *** *p* < 0.001.

**Figure 5 ijms-26-09126-f005:**
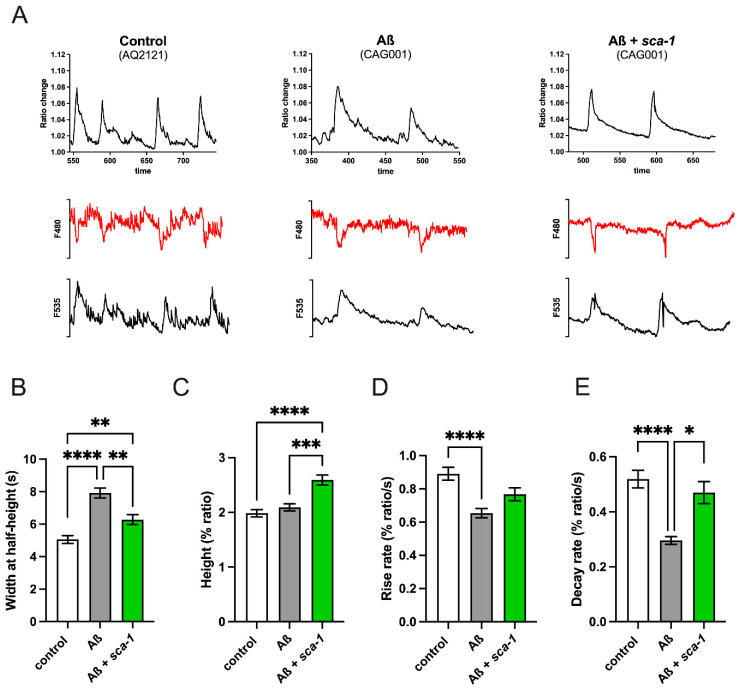
Impact of Aß overexpression and *sca-1* RNAi silencing on *C. elegans* vulva cytosolic [Ca^2+^] at day 4 of adulthood. (**A**) At the top, 200 sec of typical ratio records of spontaneous cytosolic [Ca^2+^] oscillations obtained from control worms (AQ2121 strain), Aß overexpressing worms (CAG001 strain) and Aß overexpressing worms + *sca-1* 100% RNAi. At the bottom are represented the single wavelength records with emission at 480 and 535 nm corresponding to the 200 s ratio records of the panels above, showing the expected mirror changes. Panels (**B**–**E**) show the mean peak parameters of the vulva cytosolic [Ca^2+^] oscillations, as: (**B**) Width at half-height, expressed as the time necessary to reach the half maximum height of the calcium peak. (**C**) Height of calcium peak, represented as the fraction of height calcium peak change. (**D**) Rise rate, calculated as height of calcium peak divided by time to peak. (**E**) Decay rate, calculated as height of calcium peak divided by decay time. All experiments are the result of three biological replicates, each containing almost 3 technical replicates (n = 68–226 peaks analyzed). Data are mean ± s.e.m. * *p* < 0.05; ** *p* < 0.01; *** *p* < 0.001; **** *p* < 0.0001.

## Data Availability

The raw data supporting the conclusions of this manuscript will be made available by the authors, without undue reservation, to any qualified researcher.

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
