# Peer review of "SERCA Silencing Alleviates Aß(1-42)-Induced Toxicity in a C. elegans Model"

_ijms, 2025, doi:10.3390/ijms26189126_

Round 1
Reviewer 1 Report
Comments and Suggestions for Authors This manuscript represents a breakthrough article about the sarco-endoplasmic reticulum Ca²⁺-ATPase (SERCA) silencing alleviates Aß1-42-induced toxicity in a C. elegans model.Regarding the introduction I find interesting to consider C. elegans as a valuable organism for modeling AD and studying the contribution of the Ca2+ signaling machinery to AD pathogenesis. The C. elegans genome contains sca-1, a highly conserved orthologue of the mammalian SERCA. sca-1 is primarily expressed in contractile tissues, including body-wall muscle cells. Lack of sca-1 results in muscle dysfunction and in early larval or embryonic lethality, indicating that its activity is required for embryonic development.
The authors have previously reported that both pharmacological inhibition and RNAi-induced silencing of sca-1 in post-developed individuals promote health-span and longevity, suggesting that submaximal inhibition of sca-1 triggers adaptations that improve organism survival in adult worms.
But in neurodegeneration paradigms in C. elegans, the role of sca-1 remains under discussion. Several studies reported that inhibition of sca-1 activity with thapsigargin potentiated neurodegeneration. Conversely, sca-1 inhibition with CPA in an α-synuclein overexpression model in dopaminergic neurons and sca-1 inhibition with thapsigargin in a model of Aβ overexpression in glutamatergic neurons prevented neuro-degeneration. These results indicate that sca-1 inhibition might be protective in models of toxic protein aggregation, although the underlying mechanisms are largely unknown. To get new insights into the influence of SERCA activity in the pathogenesis of AD, here we explore for the first time the impact of RNAi-driven sca-1 silencing on Aβ-induced toxicity in a C. elegans model that overexpresses the most amyloidogenic version 85 of the human Aβ1-42, in body-wall muscle cells. The Sarco Endoplasmic Reticulum Ca2+-ATPase (SERCA) pumps cytosolic Ca2+ into the endoplasmic reticulum lumen (ER) to maintain cytosolic and ER Ca2+ levels under physiological conditions. Previous reports suggest that cellular Ca2+ homeostasis is compromised in Alzheimer`s Disease (AD) and that SERCA activity can modulate the phenotype of AD mouse models. Here, we used a C. elegans strain that overexpresses the most toxic human ß-amyloid peptide (Aß1-42) in body-wall muscle cells to study the effects of SERCA (sca-1) silencing on Aß1-42-induced body-wall muscle dysfunction. sca-1 knockdown reduced the percentage of paralyzed worms, improved locomotion in free-mobility assays, and restored pharynx pumping in Aß1-42-overexpressing worms. At the cellular level, sca-1 silencing partially prevented Aß1-42-induced exacerbated mitochondrial respiration and mitochondrial ROS production, and restored mitochondrial organization around sarcomeres. sca-1 knockdown reduced the number and size of Aß1-42 aggregates in body-wall muscle cells and prevented the formation of Aß1-42 oligomers. Aß1-42 expression induced slower kinetics of spontaneous cytosolic Ca2+ transients in muscle cells and sca-1 partially restored these changes. We propose that partial sca-1 loss of function prevents the toxicity associated with beta-amyloid accumulation by reducing the formation of Aß1-42 oligomers and improving mitochondrial function, in a mechanism that requires remodeling of cytosolic Ca2+ dynamics and partial ER Ca2+ depletion. To further characterize the impact of sca-1 silencing on Aß(1-42)-induced body-wall muscle dysfunction, we measured the impact of sca-1 silencing on locomotion parameters through free-mobility assays. Locomotion parameters measured included distance traveled and speed of movement, either as maximum speed, average speed, or speed measured in body lengths per second (BLPS) to account for differences in worm size. As previously reported [27], our results show that Aß(1-42)-overexpressing worms exposed to temperature challenge significantly reduced all mobility parameters during aging compared 112 to N2 control worms (Fig.1E,G,I,K). Regarding results, of the article disruptions in intracellular calcium (Ca²⁺) homeostasis have been consistently linked to the pathogenesis of Alzheimer’s disease (AD). Among the diverse systems regulating intracellular Ca²⁺ dynamics, the activity of the SERCA has been shown to modulate phenotypic outcomes in pre-clinical AD models, although the cellular mechanisms underlying its influence remain poorly defined. Here, we show that RNAi-mediated silencing of sca-1, which encodes the C. elegans ortholog of SERCA, partially ameliorates Aβ(1–42)-induced phenotypes, including paralysis, impaired locomotion, and reduced pharyngeal pumping (Fig. 1). These findings suggest that partial inhibition of sca-1, and thus a reduced capacity for ER Ca²⁺ sequestration, confers protection against Aβ(1–42)-mediated toxicity. Furthermore, overexpression of Aβ(1–42) in body-wall muscle cells was found to perturb cytosolic Ca²⁺ transient dynamics without significantly affecting basal cytosolic Ca²⁺ levels. Notably, sca-1 silencing partially restored the dynamics of cytosolic Ca2+ transients (Fig. 5), suggesting that its protective effect may stem from both compensatory responses of alternative Ca2+ transport systems and a reduction in the ER Ca2+ level. This interpretation is further substantiated by our previous findings that crt-1 silencing, which disrupts the ER major Ca²⁺ buffering mechanism, also prevents Aß(1-42)-induced body-wall muscle dysfunction, and the lack of synergistic effect on mobility of upon double sca-1/crt-1 silencing (Suppl.Fig.2). Collectively, these results highlight the pathological role of excessive ER Ca²⁺ accumulation in Aβ toxicity and suggest that targeting ER Ca²⁺ homeostasis may represent a potential therapeutic strategy in AD. To elucidate the mechanisms underlying the protective effect of sca-1 on muscle function, the authors demonstrate that sca-1 silencing is associated with reduced levels of Aβ(1-42) aggregates, enhanced mitochondrial network integrity in body-wall muscle cells, and decreased Aβ(1-42)-induced superoxide production in pharyngeal mitochondria. Activation of the mPTP disrupts the inner mitochondrial membrane ion, metabolite and electrical gradients, leading to mitochondrial dysfunction and the release of pro-apoptotic factors . The authors’ findings indicate that Aβ(1-42) overexpression induces mitochondrial stress, as evidenced by elevated superoxide levels and increased basal oxygen consumption (Fig.2). Under these conditions, mitochondria become more susceptible to mPTP opening in response to Ca²⁺ overload. We hypothesize that sca-1 silencing, which leads to reduced ER Ca²⁺ levels, diminishes ER-to-mitochondria Ca²⁺ transfer. As a result, mitochondria are less prone to mPTP opening, thereby maintaining functional competence under stress conditions. Moreover, we have previously shown that submaximal sca-1 silencing in post-developed C. elegans promotes lifespan and health span [20]. These findings suggest that the pro-survival effect is mediated by a reduced ER-to-mitochondria Ca²⁺ transfer and requires activation of AMP-activated protein kinase (AMPK) alongside inhibition of the mechanistic target of rapamycin (mTOR) pathway. Based on this evidence, it is plausible to hypothesize that a metabolic shift favoring catabolic pathways while repressing anabolic activity may compensate for Aβ(1-42)-induced mitochondrial dysfunction, thereby contributing to the protective phenotype observed in the model of the authors.
Our results demonstrate that overexpression of Aβ(1-42) in body-wall muscles significantly attenuates the kinetics of spontaneous cytosolic Ca²⁺ transients in muscle cells, a 369 phenotype that is partially rescued by sca-1 knockdown (Fig.5). For obtaining that results, regarding the methods used, the authors used smart way: 4.1. C. elegans strains and maintenance. 4.2. RNAi feeding worms and experimental specifications. 4.3. RNA extraction. 4.4. cDNA synthesis and qRT-PCR. 4.5. Paralysis assays. 4.6. Tracking assays. 4.7. Superoxide measurements. 4.9. Body-wall muscle mitochondrial organization. 4.10. Oxygen-consumption assays. 4.11. Calcium measurements. 4.12. Aß in vivo Aggregate Staining. 4.13. Protein Extraction and Wester Blotting. Statistical analyses were performed using Prism software (GraphPad). All data sets were tested for normal distribution using the Kolmogorov-Smirnov test. Unless otherwise indicated, significance between two experimental groups was determined using a two-tailed unpaired Student's t-test whereas group sets were analyzed using a one-way ANOVA (post-hoc Tukey test). NS, not significant; *p < 0.05; **p < 0.01; ***p < 0.001; ****p < 0.0001.
This article is well-written and -elaborated, and deserves to be published in its present form.
Congratulations.
Author Response
We're glad you like it.
Reviewer 2 Report
Comments and Suggestions for Authors
The report by Caldero-Escudero and coworkers describe an investigation to determine effects of sca-1 knockdown on Abeta pathologies using a transgenic C. elegans model. The study hopes to increase understanding of the role the Sarco Endoplasmic Reticulum Ca2+-ATPase (SERCA) pump in mediating amyloid toxicity. The study observes improvements in multiple measures amyloidosis including paralysis, locomotion, development, and protein aggregation both in vitro and in vivo. The study is well designed considering the sca-1 gene null mutant is lethal and thus only knock-down approaches are available. The authors observations add to the knowledge of SERCA functions in proteopathies, especially those involving amyloid beta. A few minor suggestions should clarify a few points.
- The authors should also conduct sca-1 RNAi on N2 only. This, as a control, should rule out that the improved effects on locomotion are not simply a movement stimulus effect unrelated to amyloid.
- Alzheimer’s disease pathophysiology principally occurs in the brain. Authors should discuss the limitations of their conclusions based upon muscle cell expression of the GMC101 strain.
- Considering the relatively weak RNAi effect (~50%), one wonders why authors did not cross GMC101 with other strains such as rrf-3 or sid-1 to enhance the RNAi effect.
- As the SERCA gene is encoded by 3 genes in humans (ATP2A1, ATP2A2, and ATP2A3) one wonders about their expression levels during aging and during Alzheimer’s disease ?
- The role of SERCA in neurodegeneration has been explored previously. The authors should articulate the advances and new knowledge provided in the nematode C. elegans in the discussion section.
Author Response
The report by Caldero-Escudero and coworkers describe an investigation to determine effects of sca-1 knockdown on Abeta pathologies using a transgenic C. elegans model. The study hopes to increase understanding of the role the Sarco Endoplasmic Reticulum Ca2+-ATPase (SERCA) pump in mediating amyloid toxicity. The study observes improvements in multiple measures amyloidosis including paralysis, locomotion, development, and protein aggregation both in vitro and in vivo. The study is well designed considering the sca-1 gene null mutant is lethal and thus only knock-down approaches are available. The authors observations add to the knowledge of SERCA functions in proteopathies, especially those involving amyloid beta. A few minor suggestions should clarify a few points.
1.The authors should also conduct sca-1 RNAi on N2 only. This, as a control, should rule out that the improved effects on locomotion are not simply a movement stimulus effect unrelated to amyloid.
We thank the reviewer for this excellent question, we actually tested the effect of sca-1 RNAi on locomotion (Suppl. Fig. 1) in N2 worms and we observed that it also improves mobility. We mention: “Off note, sca-1 silencing in wild-type nematodes (Suppl.Fig1A) also improves mobility parameters, although the major effect is observed in young adults (Suppl.Fig.1B-E). This observation points out that mobility improvement driven by sca-1 knock-down does not rely on Aß(1-42)-overexpression.”
2.Alzheimer’s disease pathophysiology principally occurs in the brain. Authors should discuss the limitations of their conclusions based upon muscle cell expression of the GMC101 strain.
We agree with the reviewer, C. elegans models with Aβ expressed in muscle are great for studying aggregation and general proteotoxicity, but the model does not capture neuron-specific processes such as synaptic dysfunction, neurotransmission changes, or neuronal cell death. We have included the following sentence in the discussion section:
“Transgenic C. elegans models that ectopically overexpress Aβ in body-wall muscle cells constitute a robust system for investigating mechanisms of protein aggregation and proteotoxic stress. However, their translational relevance to Alzheimer’s disease remains limited, as they do not reproduce fundamental aspects of the disorder’s neuropathophysiology, including synaptic dysfunction, dysregulated neurotransmission, and progressive neuronal degeneration.”
3.Considering the relatively weak RNAi effect (~50%), one wonders why authors did not cross GMC101 with other strains such as rrf-3 or sid-1 to enhance the RNAi effect.
We thank the reviewer for the proposal, and we will certainly use the approach suggested to enhance the efficiency of sca-1 RNAi in follow up research.
4.As the SERCA gene is encoded by 3 genes in humans (ATP2A1, ATP2A2, and ATP2A3) one wonders about their expression levels during aging and during Alzheimer’s disease?
We build a table summarizing how the expression of different human SERCA isoforms respond to ageing in muscle and brain. The best reported influence of aging is decreased ATP2A2 (SERCA2) in aged human heart.
|
Tissue (human) |
ATP2A1 (SERCA1) |
ATP2A2 (SERCA2) |
ATP2A3 (SERCA3) |
Summary / Notes |
Key references (PMID) |
|
Heart (left ventricle / myocardium) |
Not a major cardiac isoform — no clear aging signal |
Decrease with age (robust: mRNA/protein/activity decline; linked to impaired relaxation/diastolic dysfunction) |
Low/variable in cardiomyocytes; no clear aging consensus |
Strongest and most reproducible signal: ATP2A2 decreases with age in human heart |
Cain et al., JACC 1998 (PMID: 9737529); Bers et al., Cardiovasc Res 2002 review (PMID: 11884269); Volkova & Smaili 2005 review (PMID: 15923321) |
|
Skeletal muscle (bulk adult muscle) |
Mixed / no consistent change across cohorts |
Mixed / no consistent change; functional SERCA activity declines, but mRNA direction varies |
Low/variable; insufficient aging data |
Human muscle meta-analyses show many age genes, but ATP2A1/2 not consistently among them |
Su et al., Skeletal Muscle 2015 (PMID: 26698180); Kurochkina et al., Aging Cell 2024 (PMID: 39531906) |
|
Brain (multiple regions) |
Low expression; no consistent aging signal |
Broadly expressed (SERCA2b); no reproducible age-direction |
Expressed in subsets; no consensus |
GTEx analyses: age affects many genes, but ATP2As not consistently altered across brain |
GTEx Consortium, Cell 2017 (PMID: 29022597); Yamamoto et al., 2022 tissue-specific aging impacts (PMID: 35879532) |
Regarding the impact of Alzheimer’s disease on SERCA isoforms, we have not found evidence for a reproducible ATP2A1, ATP2A2 and ATP2A3 increase or decrease in human AD brains (PMIDs: 32668255, 32985496, 32284590).
5.The role of SERCA in neurodegeneration has been explored previously. The authors should articulate the advances and new knowledge provided in the nematode C. elegans in the discussion section.
We thank the reviewer for raising this point, but we think that in the introduction section we already summarize previous research addressing the importance of SERCA function in C. elegans neurodegeneration to put our new findings in context.
“In the context of neurodegeneration paradigms in C. elegans, the role of sca-1 remains under discussion. Several studies reported that inhibition of sca-1 activity with thapsigargin potentiated neurodegeneration [22,23]. Conversely, sca-1 inhibition with CPA in an α-synuclein overexpression model in dopaminergic neurons and sca-1 inhibition with thapsigargin in a model of Aβ overexpression in glutamatergic neurons prevented neurodegeneration [24,25]. These results indicate that sca-1 inhibition might be protective in models of toxic protein aggregation, although the underlying mechanisms are largely unknown.”
Reviewer 3 Report
Comments and Suggestions for Authors
This paper reports an analysis of Alzheimer’s disease using C. elegans as a model. The study utilizes a strain that overexpresses the highly toxic human-derived Aβ1-42 in the body wall muscles of C. elegans and evaluates the effects of knocking down the SERCA homolog gene sca-1 by RNAi, which is particularly interesting. Additionally, it suggests the potential for clinical application, indicating the significant value of this paper. On the other hand, there are several points to note.
- In this study, we use a model in which human-derived Aβ1-42 is overexpressed in the body wall muscles of C. elegans. However, there is insufficient discussion regarding the extent to which this model can replicate the neurodegeneration and brain Aβ accumulation observed in human AD pathology. The limitations of extrapolating phenotypes from muscle cells to neurodegenerative disease models should be clearly stated.
- sca-1 is a homolog of SERCA, but it is necessary to verify whether the effects of the knockdown are not due to off-target effects or changes in compensatory Ca²⁺ handling pathways. I think including confirmation of RNAi specificity or gene rescue experiments would make the findings more convincing. What do you think?
- The hypothesis is that sca-1 knockdown leads to a series of mechanisms: "suppression of Aβ oligomer formation → improvement of mitochondrial function → behavioral improvement." However, at present, this remains at the level of correlation, and there is a lack of experiments demonstrating direct causality. In particular, it is necessary to explain the molecular mechanisms mediating between "changes in ER Ca²⁺ dynamics" and "suppression of Aβ oligomer formation."
- It is unclear which fluorescent indicators and time-series analyses were used to measure the Ca²⁺ transients. I think it would be better to add a description of the quantitative analysis methods.
- This study suggests SERCA as a therapeutic target for AD, but it remains unclear whether "partial reduction of SERCA function" would be beneficial in neurons as well. When discussing the potential for clinical application, it is important to also consider possible side effects, such as the risk of ER stress and heightened UPR activity.
- The "Fig.." notation in the upper left of each figure is not necessary.
- An explanation is needed as to whether the top and bottom photos in Fig2K and Fig4A are different items or the same type.
- I believe that writing the conclusion as a separate section can appeal more to the readers.
Author Response
This paper reports an analysis of Alzheimer’s disease using C. elegans as a model. The study utilizes a strain that overexpresses the highly toxic human-derived Aβ1-42 in the body wall muscles of C. elegans and evaluates the effects of knocking down the SERCA homolog gene sca-1 by RNAi, which is particularly interesting. Additionally, it suggests the potential for clinical application, indicating the significant value of this paper. On the other hand, there are several points to note.
1.In this study, we use a model in which human-derived Aβ1-42 is overexpressed in the body wall muscles of C. elegans. However, there is insufficient discussion regarding the extent to which this model can replicate the neurodegeneration and brain Aβ accumulation observed in human AD pathology. The limitations of extrapolating phenotypes from muscle cells to neurodegenerative disease models should be clearly stated.
We agree with the reviewer, C. elegans models with Aβ expressed in muscle are great for studying aggregation and general proteotoxicity, but the model does not capture neuron-specific processes such as synaptic dysfunction, neurotransmission changes, or neuronal cell death. We have included the following sentence in the discussion section:
“Transgenic C. elegans models that ectopically overexpress Aβ in body-wall muscle cells constitute a robust system for investigating mechanisms of protein aggregation and proteotoxic stress. However, their translational relevance to Alzheimer’s disease remains limited, as they do not reproduce fundamental aspects of the disorder’s neuropathophysiology, including synaptic dysfunction, dysregulated neurotransmission, and progressive neuronal degeneration.”
2.sca-1 is a homolog of SERCA, but it is necessary to verify whether the effects of the knockdown are not due to off-target effects or changes in compensatory Ca²⁺ handling pathways. I think including confirmation of RNAi specificity or gene rescue experiments would make the findings more convincing. What do you think?
We thank the reviewer for the comment. The experiment proposed would certainly contribute to confirm the specificity of the phenotype observed upon sca-1 silencing. Actually, it would be an easy experiment using sca-1 mutant strains with reduced pumping activity; however, our RNAi settings makes difficult the approach proposed:
1- In our manuscript we have used an Arhinger´s library strain (HT115) to induce gene knockdown. HT115 strains contain an L4440 plasmid that express the full-length double stranded mRNA of the gene under the control of two T7 flanking bidirectional promoters to induce the knock down. As every region of the sca-1 mRNA can be targeted by the RNA-Induced Silencing Complex (RISC), we cannot reintroduce in a rescue experiment sca-1 versions lacking the RNAi target sites. This technical issue can be circumvented by using an RNAi targeting the sca-1 UTR regions, however we do not have neither the experience nor the capacity to generate this new RNAi.
2- The introduced transgene must produce enough of the functional protein to counteract the effects of the gene knockdown. To find a proper balance between RNAi-driven sca-1 degradation and sca-1 overexpression is challenging.
3- Transgene overexpression usually requires a microinjection system that we have not yet functionally operating in our laboratory.
3.The hypothesis is that sca-1 knockdown leads to a series of mechanisms: "suppression of Aβ oligomer formation → improvement of mitochondrial function → behavioral improvement." However, at present, this remains at the level of correlation, and there is a lack of experiments demonstrating direct causality. In particular, it is necessary to explain the molecular mechanisms mediating between "changes in ER Ca²⁺ dynamics" and "suppression of Aβ oligomer formation."
This is a fair question. Although we have not captured the molecular mechanisms linking the regulation of SERCA with the suppression of the Aβ oligomer formation we provide full paragraph in the discussion section addressing this topic. New experiments designed to answer this question would be definitely explored in a follow up paper.
“Another important finding of this manuscript is the reduction of Aβ(1-42) aggregates and high molecular weight Aβ(1-42) oligomers upon sca-1 silencing. We noticed that our new results contrast with those previously reported, indicating sca-1 silencing did not impact on the levels of Aβ(1-42) in the same C. elegans strain we have used here [28]. The discrepancy may be explained by the use of an RNAi dose 10 times lower in those experiments. Given the established toxicity of oligomeric Aß species, reduced Aβ(1-42) accumulation might therefore explain per se the improvement in the parameters of body-wall muscle functionality observed in GMC101 sca-1 knockdown worms (Fig.1). In our experimental setting, Aβ(1-42) steady state levels result from a balance between transcriptional overproduction of Aβ(1-42) driven by the muscle-specific promoter unc-54/myo-4 and the action of Aβ(1-42) clearance mechanisms. Since sca-1 silencing did not impact on Aβ(1-42) mRNA levels, it can be hypothesized that lack of sca-1 may reduce Aβ(1-42)-deposits by activating protein clearance mechanisms. Mechanisms previously associated to Aβ-clearance include ubiquitin-proteosome degradation, chaperone-mediated autophagy, or macro-autophagy [40]. A mechanistic interpretation of this phenomenon may be hypothesized. Partial sca-1 loss-of-function elicits endoplasmic reticulum (ER) stress, thereby activating the unfolded protein response (UPR) and inducing transcriptional upregulation of chaperones, including hsp-4 in C. elegans [41–43]. The UPR signaling cascade facilitates the induction of macroautophagy, which serves as a proteostatic mechanism for the degradation of misfolded or aggregated proteins [44]. Importantly, amyloid-β (Aβ) has been demonstrated to be a substrate for autophagy-mediated lysosomal degradation in both rodent models and C. elegans [44,45]. Collectively, we propose that sca-1 RNAi-driven UPR activation, coupled with enhanced chaperone expression and autophagic flux, may potentiate the catabolism of Aβ species, thereby promoting the clearance of proteotoxic aggregates and the re-establishment of intracellular proteostasis.”
4.It is unclear which fluorescent indicators and time-series analyses were used to measure the Ca²⁺ transients. I think it would be better to add a description of the quantitative analysis methods.
We thank the reviewer for the comment.
We have now included in the methods section a new sentence clarifying the fluorescent indicator used: “Body-wall muscle cytosolic [Ca2+] ([Ca2+]cyt) measurements were performed using the strains AQ2121 and CAG001. Both expresses the ratiometric Ca2+ sensor YC2.1 in body wall muscle cells cytosol.“
We have also included in the methods section a brief description of the quantitative analysis method used to describe the Ca2+ recordings: “Then, the fluorescences obtained at 480 and 535 nm emission were analyzed off-line using a home-made specific algorithm designed to first identify the Ca2+ peaks by looking for mirror changes in both wavelengths and then to calculate for each Ca2+ peak in each experiment the total width and the width at mid-height expressed in seconds, the height obtained as a percentage of ratio change from the base to the peak, the time to rise from the base to the peak and the time to return to the base from the peak.”
5.This study suggests SERCA as a therapeutic target for AD, but it remains unclear whether "partial reduction of SERCA function" would be beneficial in neurons as well. When discussing the potential for clinical application, it is important to also consider possible side effects, such as the risk of ER stress and heightened UPR activity.
We thank the reviewer for the comment. We have now updated the last paragraph of the discussion rising the translatability issue of our findings on muscle cells to neurons.
“However, their translational relevance to Alzheimer’s disease remains limited, as they do not reproduce fundamental aspects of the disorder’s neuropathophysiology, including synaptic dysfunction, dysregulated neurotransmission, and progressive neuronal degeneration. Interestingly, brain-specific partial loss of SERCA2 (ATP2A2) function has been reported to cause only mild behavioral abnormalities [43]. However, its potential protective role against pathology in mouse models of Alzheimer’s disease remains unexplored.”
6.The "Fig.." notation in the upper left of each figure is not necessary.
We thank the reviewer for pointing it out. We send new figures lacking the Fig. notation.
7.An explanation is needed as to whether the top and bottom photos in Fig2K and Fig4A are different items or the same type.
We are not sure if we fully understand the comment. These are different pictures taken from worms in different experiments.
8.I believe that writing the conclusion as a separate section can appeal more to the readers.
We thank the reviewer for the suggestion. We have now generated a new section with the main conclusions of the manuscript.
“CONCLUSIONS
We conclude that partial loss of sca-1 function enhances muscle performance in a C. elegans model of proteotoxicity characterized by Aβ(1–42) accumulation in body-wall muscle cells. This improvement is associated with reduced levels of Aβ(1–42) oligomers and aggregate formation, enhanced mitochondrial function, and the restoration of cytosolic Ca²⁺ transient kinetics. These observations support a model in which sca-1 knockdown mitigates β-amyloid–induced toxicity by limiting Aβ(1–42) oligomerization and promoting mitochondrial function through a mechanism that involves remodeling of cytosolic Ca²⁺ dynamics and partial ER Ca²⁺ depletion.”
Round 2
Reviewer 2 Report
Comments and Suggestions for Authors
The authors have satisfactorily responded to my comments.
Reviewer 3 Report
Comments and Suggestions for Authors
The authors have made appropriate corrections and additions to my previous questions and comments. The figures have also been updated, which I believe will be beneficial to readers. Some parts (5 pages) require proofreading, but this should be easily corrected. I hope the book will be published soon.